# Treatment of VLCAD-Deficient Patient Fibroblasts with Peroxisome Proliferator-Activated Receptor δ Agonist Improves Cellular Bioenergetics

**DOI:** 10.3390/cells11172635

**Published:** 2022-08-24

**Authors:** Olivia M. D’Annibale, Yu Leng Phua, Clinton Van’t Land, Anuradha Karunanidhi, Alejandro Dorenbaum, Al-Walid Mohsen, Jerry Vockley

**Affiliations:** 1Division of Genetic and Genomic Medicine, Department of Pediatrics, University of Pittsburgh School of Medicine, Pittsburgh, PA 15224, USA; 2Department of Human Genetics, University of Pittsburgh Graduate School of Public Health, Pittsburgh, PA 15261, USA; 3Reneo Pharmaceuticals, Inc., 18575 Jamboree Road Suite 275-S, Irvine, CA 92612, USA; 4UPMC Children’s Hospital of Pittsburgh, Pittsburgh, PA 15224, USA

**Keywords:** fatty acid oxidation, VLCAD deficiency, ACADs, PPARs, acyl-CoA dehydrogenases, REN001, cellular bioenergetics

## Abstract

**Background:** Very long-chain acyl-CoA dehydrogenase (VLCAD) deficiency is an autosomal recessive disease that prevents the body from utilizing long-chain fatty acids for energy, most needed during stress and fasting. Symptoms can appear from infancy through childhood and adolescence or early adulthood, and include hypoglycemia, recurrent rhabdomyolysis, myopathy, hepatopathy, and cardiomyopathy. REN001 is a peroxisome-proliferator-activated receptor delta (PPARδ) agonist that modulates the expression of the genes coding for fatty acid β-oxidation enzymes and proteins involved in oxidative phosphorylation. Here, we assessed the effect of REN001 on VLCAD-deficient patient fibroblasts. **Methods:** VLCAD-deficient patient and control fibroblasts were treated with REN001. Cells were harvested for gene expression analysis, protein content, VLCAD enzyme activity, cellular bioenergetics, and ATP production. **Results:** VLCAD-deficient cell lines responded differently to REN001 based on genotype. All cells had statistically significant increases in *ACADVL* gene expression. Small increases in VLCAD protein and enzyme activity were observed and were cell-line- and dose-dependent. Even with these small increases, cellular bioenergetics improved in all cell lines in the presence of REN001, as demonstrated by the oxygen consumption rate and ATP production. VLCAD-deficient cell lines containing missense mutations responded better to REN001 treatment than one containing a duplication mutation in *ACADVL*. **Discussion:** Treating VLCAD-deficient fibroblasts with the REN001 PPARδ agonist results in an increase in VLCAD protein and enzyme activity, and a decrease in cellular stress. These results establish REN001 as a potential therapy for VLCADD as enhanced expression may provide a therapeutic increase in total VLCAD activity, but suggest the need for mutation-specific treatment augmented by other treatment measures.

## 1. Introduction

Long-chain fatty acids enter cells via protein fatty acid transporters on the cell surface concurrent with or followed by conjugation to a CoA group by a fatty acyl-CoA synthases (FACS) [1,2]. Long-chain fats are activated in the cytoplasm and require a series of three enzymatic steps that constitute what is known as the carnitine cycle [1,2]. Carnitine palmitoyl transferase 1 (CPT1) replaces the CoA moiety of the long-chain acyl-CoA with carnitine (acylcarnitine), which is transported by carnitine-acylcarnitine translocase (CAT) across the inner mitochondrial membrane in exchange for a free carnitine molecule from the mitochondrial matrix [1,2]. The carnitine of the acylcarnitine is replaced with a CoA and is released as an acyl-CoA ester by carnitine palmitoyl transferase 2 (CPT2), where it can then enter the fatty acid β-oxidation pathway, a series of four enzymatic steps that results in the production of a two carbon acetyl-CoA, one NADH, and one FADH_2_, regenerating an acyl-CoA that is now two carbons shorter [1,2,3]. Very long-chain acyl-CoA dehydrogenase (VLCAD) catalyzes the α,β-dehydrogenation of long-chain acyl-CoA substrates with various carbon chain lengths and maximal activity to C14-CoA to its enoyl-CoA product, utilizing the electron transfer flavoprotein (ETF), a mitochondrial matrix electron shuttle protein, as an electron acceptor [1,2,4]. Reduced ETF transfers its reducing equivalents to its redox partner, the ETF dehydrogenase (ETFDH), which in turn delivers the reducing equivalents to the ubiquinone pool and complex III of the electron transport chain (ETC) [1,2,4]. 

VLCAD deficiency (VLCADD) is an autosomal recessive disorder caused by biallelic mutations in the *ACADVL* gene [5]. The frequency of VLCADD in various populations is between 1:30,000 and 1:100,000 live births [6,7]. Symptoms of VLCADD include hypoglycemia, recurrent rhabdomyolysis, myopathy, hepatopathy, and cardiomyopathy. Symptoms can present in infancy, later in childhood, or in adolescence to early adulthood [8]. Treatment for VLCADD patients involves a low-fat diet consisting mainly of medium-chain triglyceride (MCT) or triheptanoin supplementation with smaller, more frequent meals [9,10,11,12]. However, many patients still have episodes of rhabdomyolysis and cardiomyopathy that can lead to hospitalization, suggesting the need for additional treatment options. Episodes of metabolic decompensation are typically triggered by physiologic stress such as illness or excess exercise, but the cause often remains unidentified [13]. The ultimate outcome is improved by identification of the disorder through newborn screening [14]. 

Peroxisome proliferator-activated receptors (PPARs) are nuclear receptors that play key roles in the regulation of fatty acid β-oxidation, lipid metabolism, inflammation, and cellular growth and differentiation [15,16,17,18,19]. They are divided into three categories based on the specific promotors that they stimulate: PPARα, PPARγ, and PPARδ. Pan-PPAR agonists globally upregulate all PPARs and affect up to 25% of the genome [15,16,17,18,19]. Individually, PPARα has the broadest specificity, with involvement in the regulation of energy homeostasis, but also a myriad of other cellular activities [20,21]. Its stimulation reduces triglyceride levels and has been proposed as a target for the treatment of disorders of energy metabolism. PPARγ enhances glucose metabolism and insulin sensitization [21]. PPARδ is a major activator of oxidative metabolism and is ubiquitously expressed [15,22,23]. It is activated by polyunsaturated fatty acids such as arachidonic acid, oleic acid, dexamethasone, and eicosanoids such as prostaglandin 1 (PGA_1_), carbaprostacyclin (cPGI), and 15-deoxy-∆^12,14^-prostaglandin J_2_ (15d-J_2_) [24,25]. In vivo experiments with PPARδ agonists have examined their effects in a variety of diseases and cellular processes, including diabetes, obesity, and lipid metabolism. In a two-week clinical study, treatment of moderately obese men with dyslipidemia with GW501516, a PPARδ agonist, resulted in a decrease in fasting and postprandial plasma triglycerides, low-density lipoprotein (LDL) cholesterol, and apoB compared to placebo, as well as reductions in liver fat content and urinary isoprostanes (a marker of whole-body oxidative stress) [26]. Four weeks of treatment of insulin-resistant middle-aged obese rhesus monkeys with GW501516 induced a dose-dependent rise in serum high-density lipoprotein cholesterol while lowering the levels of small-dense LDL, fasting triglycerides, and fasting insulin [27]. Genetically obese *ob/ob* mice had reduced plasma glucose and blood insulin levels after treatment with GW501516 [22]. Genetically predisposed obese Lepr*^db/db^*mice treated with GW501516 demonstrated a decrease in lipid accumulation, while PPARδ-deficient mice were prone to obesity on a high-fat diet [19]. The PPARδ agonist HPP593 has been reported to prevent renal necrosis under chronic ischemia [28]. Recent studies with herbal supplements such as bavachinin (a pan-PPAR agonist) from the glucose-lowering malaytea scurfpea herb and ginger (a PPARδ agonist) reduced obesity in obese *db/db* mice, and diet-induced obesity in C57BL6J mice, respectively [29,30]. 

Bezafibrate is a pan-PPAR agonist used to treat hyperlipidemia as it increases high-density (HDL) cholesterol levels, decreasing total and LDL cholesterol levels [31]. Since pan-PPARs can increase fatty acid β-oxidation, there has been interest in repurposing bezafibrate as a treatment for fatty acid oxidation disorders. In an in vitro study, VLCADD patient-derived fibroblast cell lines treated with two versions of bezafibrate demonstrated a three-fold increase in palmitate oxidation with an increase in VLCAD mRNA, protein, and enzyme activity. RT-PCR showed an increase in other genes encoding proteins in the β-oxidation pathway [32]. Similarly, treatment of CPT2-deficient human myoblast cells with bezafibrate and the PPARα agonist GWδ 0742 led to an increase in *CPT1-B* and *CPT2* mRNA levels with increased CPT2 activity, while GWα 7647, another PPARα agonist, had a minimal effect [33]. Treatment with bezafibrate of fibroblasts from 26 patients with mitochondrial fatty acid oxidation trifunctional protein (MTP) deficiency with various mutations led to improved cellular palmitate oxidation in 6 of 26 cell lines [34]. In an open-label trial treating patients with CPT2 deficiency, patients showed an increase or no change in the incidence of rhabdomyolysis episodes, but an improvement in quality of life scores [35,36]. However, in a randomized, double-blind, placebo-controlled clinical trial in patients with VLCAD or CPT2 deficiency, bezafibrate failed to improve cardiac function or whole-body fatty acid oxidation [37]. One possibility for this dichotomy is the limited PPARδ effect of bezafibrate. 

Finally, in a clinical study examining the effect of resveratrol, a mitochondrial antioxidant, it had no effect on exercise tolerance or whole-body fatty acid oxidation in patients with VLCAD or CPT2 deficiency. Thus, a clinical need for additional therapies for this group of disorders remains. REN001 is currently involved in a clinical trial in the United States for FAO-deficient patients, including VLCADD (https://clinicaltrials.gov/ct2/show/NCT04482049?term=reneo&draw=2&rank=3, accessed on 22 August 2022) and primary mitochondrial myopathy (https://clinicaltrials.gov/ct2/show/NCT05267574?term=Reneo&draw=2&rank=1, accessed on 22 August 2022). In this study, we examined the effects of REN001 in VLCADD patient-derived fibroblast cell lines to determine the overall impact on cellular bioenergetics. 

REN001 (formerly known as HPP593), a PPARδ agonist (Reneo Pharmaceuticals), has been shown to reduce oxidative stress and inflammation in renovascular hypertensive Goldbatt’s 2 kidney 1 clip (2KIC) rats [30]. In particular, 2KIC mice treated with REN001 for 30 days had reduced necrosis in the kidneys, reduced oxidative-stress-responsive proteins, and decreased pro-death protein Bcl2 and adenovirus E1B 19 kDa interacting protein 3 (BNIP3) in kidney tubules [28]. When activated, BNIP3 is integrated into the mitochondrial membrane and induces the permeabilization of the mitochondria, loss of membrane potential, and activation of mitochondrial death [30]. REN001′s proposed mechanism is the inhibition of BNIP3 activation, resulting in preserved mitochondrial function and oxidative stress control [28]. 

## 2. Materials and Methods

Experiments were performed in accordance with approved local and regional guidelines and regulations. Experimental human protocols were approved by the Institutional Review Board at the University of Pittsburgh, protocol 19030195.

### 2.1. Subjects

Skin biopsies for fibroblast culture were performed on a clinical basis from patients with various mutations in *ACADVL*, with written informed consent from patients and/or parents (Table 1). Control fibroblast cells were obtained from the American Type Culture Collection (ATCC.org). 

### 2.2. Cell Culture and Treatments

Cell lines were grown in Dulbecco’s Modified Eagle Medium (DMEM, Corning Life Sciences, Manassas, VA, USA) containing high glucose levels (4.5 g/L) or in DMEM devoid of glucose for 48 h. Both media were supplemented with 10% fetal bovine serum (FBS), 4 mM L-glutamine, and 100 IU penicillin and 100 µg/mL streptomycin (Corning Life Sciences). REN001 was obtained from Reneo Pharmaceuticals, Irvine, CA, USA, and resuspended from a powder in DMSO. 

Cells were treated with REN001 at 85% confluency at the following concentrations: 0, 15, 30, 60, and 120 nM. Additional cultures were treated with 600 µM bezafibrate (Sigma Aldrich, St. Louis, MO, USA). The 0 nM treatment was given DMSO as a control for both REN001 and bezafibrate. Cultures were incubated for 48 h at 37 °C, 5% CO_2_ and were harvested for analysis. 

### 2.3. Real-Time Quantitative Polymerase Chain Reaction (qPCR)

Total RNA was isolated using the RNeasy Mini Kit (Qiagen, Valencia, CA, USA) from REN001-treated VLCAD and control fibroblasts with on-column DNaseI digestion (Qiagen). First-strand synthesis of complementary DNA (cDNA) was reverse-transcribed from 2500 ng of total RNA using the Super Vilo IV Master Mix (Qiagen). Quantitative PCR was performed with an equivalent amount of cDNA on a Bio-Rad CFX96 Real-Time PCR Instrument, with SYBR Green Master Mix (Thermo Fisher Scientific, Waltham, MA. USA). *ACADVL*, *HADHA*, *HADHB*, *ETFDH*, *UQCRC2*, and *NDUFS2* were assayed using primers were obtained from PrimerBank [38,39,40] (Appendix A). Expression levels were normalized to *TOMM20* and the data were analyzed by the 2^−∆∆*Ct*^ method [41]. 

### 2.4. PPARδ Binding Site Analysis

ChIP-seq analysis for PPARδ binding sites was performed using the publicly available dataset on the Gene Expression Omnibus (GSE 50144) (PMID: 24721177), and binding sites were identified with the use of MACS2 (PMID: 24743991). ChIP-seq peaks were visualized using the IGV viewer (PMID: 21221095), and gene ontology enrichment for PPARδ target genes and pathways was analyzed using the Cistrome-GO and GREAT GO tools (PMID: 20436461) (PMID: 31053864). For the GREAT-GO analysis, the ChIP-seq peaks were analyzed using the GRCh38 human assembly with whole genome background, and basal + extension (constitutive 10.0 kb upstream and 10.0 kb downstream, up to 10.0 kb maximum extension) with curated regulatory domains selected as the associated genomic region criteria. The output from the GREAT-GO tool generates information for the GO Biological Process and Human Phenotype. 

### 2.5. Whole Cell Lysate, Protein Concentration, SDS-PAGE Gel, and Western Blot

Cell were treated with REN001 in complete DMEM with glucose for 48 h, harvested via trypsinization, pelleted, and stored at −80 °C for Western blot analysis. Pellets were lysed with 50 µL of radioimmunoprecipitation assay (RIPA) buffer (Thermo Fisher Scientific) with 1× Protease Inhibitor Cocktail (PI) (Roche, St. Louis, MO, USA) for 30 min on ice and centrifuged at 14,000× *g* for 15 min at 4 °C. Supernatants were collected and 25 µg of protein was loaded onto a 4–15% gradient Criterion precast SDS-PAGE gel (Bio-Rad, Hercules, CA, USA). Following electrophoresis, the gel was blotted onto a nitrocellulose membrane and incubated with anti-VLCAD (VLCAD 1:1000, rabbit, Vockley lab, [42]. then incubated with secondary goat anti-rabbit-HRP antibody (1:3000, BioRad). The Pierce ECL Western Blotting Substrate Kit (Thermo Fisher Scientific, Waltham, MA, USA) was used to visualize bands. Membranes were stripped and re-probed with TFP cocktail antibody (1:1000, rabbit, Vockley lab, [42]) containing antibodies for both the alpha and beta subunits, and with mouse-anti-glyceraldehyde 3-phosphate dehydrogenase (GAPDH; 1:25,000) monoclonal antibody (ABCAM, Cambridge, MA, USA) to verify equal loading. ImageLab software was used to quantify band intensity and bands were normalized to GAPDH intensity. 

### 2.6. Immunofluorescence Microscopy

Treated cultured fibroblasts were seeded at a concentration of 5 × 10^4^ cells/mL on Poly-L-Lysine-coated glass cover slips in a 12-well plate and allowed to grow overnight in growth media at 37 °C in a 5% CO_2_ incubator. Cells were then fixed in 4% paraformaldehyde for 10 min and permeabilized with 0.1% Triton X-100 and blocked after brief washings in 5% donkey serum at room temperature for 1 h. Cells were briefly washed and treated with primary antibodies VLCAD (1:1000, Vockley Lab) and HADHA (1:100, Santa Cruz Biotechnology, Dallas, TX, USA) overnight at 4 °C. After brief washing with 1 X tris buffered saline, pH 7.4 with Tween 20, cells were incubated with the secondary antibodies donkey anti-rabbit Alexa Fluor 488 and donkey anti-mouse Alexa Fluor 594 (1:1000, Invitrogen, Waltham, MA, USA) for 1 h at RT. Nuclei were counterestained with NucBlue Fixed Cell ReadyProbes Reagent (DAPI; Invitrogen). The cover slips were then mounted using mounting media before imaging. All images were taken on a Zeiss LSM710 Confocal microscope using 63× magnification. Images were analyzed using ImageJ [43]. 

### 2.7. Electron Transfer Flavoprotein (ETF) Fluorometric Reduction Assay

The ETF reduction assay was performed using a Jasco FP-6300 spectrofluorometer (Easton, Talbot County, MD, USA) with a cuvette holder heated with circulating water at 32 °C, as previously described [44]. Briefly, treated cell pellets were lysed using 50 mM Tris, pH 8.0 buffer, and 0.1 × protease inhibitor EDTA-free and sonicated twice in an ice-cold water bath sonicator at amplitude 45 for 1.5 min with 15 sec intervals. The assay was otherwise performed as described, at the indicated substrate concentrations [44]. The enzyme was diluted 1200-fold into buffer containing 50 mM Tris, pH 8.0, 5 mM EDTA, and 50% glycerol, and 10 μL was used for each assay. The ETF concentration was 2 µM. Spectra Manager 2 software (Jasco, Inc., Talbot County, MD, USA) was used to collect data and calculate reaction rates, and Microsoft Excel was used to calculate the kinetic parameters. 

### 2.8. Fatty Acid Oxidation (FAO) Flux Analysis

The tritium release assay was performed as previously described, with the noted changes [45]. Cells were grown in T175 flasks and seeded at 350,000 cells per well in 6-well plates in triplicate and in duplicate wells for protein concentration for normalization, and grown for 24 h in complete DMEM. Wells were treated with REN001 in complete DMEM for 48 h in a 37 °C/5% CO_2_ incubator. Cells were washed once with PBS and incubated with 0.34 µCi [9,10-^3^H] oleate (45.5Ci/mmol; Perkin Elmer, Waltham, MA, USA) in 50 nmol of oleate prepared in 0.5 mL glucose-free DMEM with 1 µg/mL L-carnitine and 2 mg/mL alpha-cyclodextrin for 2 h at 37 °C. Fatty acids were solubilized with alpha-cyclodextrin as described [46]. After incubation, ^3^H_2_O released was separated from the oleate on a column containing 750 µL of anion exchange resin (AG 1 × 8, acetate, 100–200 Mesh, BioRad) prepared in water. After the incubation medium was passed through the column, the plate was washed with 1 mL of water, which was also transferred to the column, and resin was washed with 1 mL of water. All eluates were collected in a scintillation vial and mixed with 10 mL of scintillation fluid (Eco-lite, MP), followed by counting in a Beckman scintillation counter in the tritium window. Standards contained a 10 µL aliquot of the incubation mix with 3 mL of deionized water and 10 mL of scintillation fluid.

### 2.9. Measurement of Mitochondrial Respiration

Oxygen consumption rate (OCR) was measured with a Seahorse XFe96 Extracellular Flux Analyzer Cell Mito Stress Test Kit (Agilent Technologies, Santa Clara, CA, USA). Fibroblasts were treated with REN001or bezafibrate resuspended in DMSO in DMEM without glucose for 48 h in 37 °C/5% CO_2_ incubator. Fibroblasts were harvested and seeded at a density of 60,000 cells per well in a 96-well seahorse plate coated in poly-D-lysine on the day of assay. The plate was centrifuged at 300 rpm for 1 min, rotated, and centrifuged again at the same settings. Cells were incubated for 1 h at 37 °C in a non-CO_2_ incubator in buffered Seahorse XF Assay Media (Agilent Technologies) and supplemented with 1 mM sodium pyruvate and 2 mM L-glutamine. Manufacturer’s directions were otherwise followed for the XF Mito Stress Test Kit (Agilent Technologies).

### 2.10. Measurement of ATP Production

Glycolytic and mitochondrial ATP production was measured with a Seahorse XF^e^96 Extracellular Flux Analyzer with an XF Real-Time ATP Rate Assay Kit (Agilent Technologies). Fibroblasts were seeded at 40,000 cells per well in complete DMEM and grown overnight in a 37 °C/5% CO_2_ incubator. Growth medium was removed and fibroblasts were treated with REN001 in complete DMEM for 48 h. Cells were washed twice with Seahorse XF Aassy Media and incubated for 1 h at 37 °C in a non-CO_2_ incubator in buffered Seahorse XF Assay Media (Agilent Technologies) supplemented with 1 mM sodium pyruvate, 2 mM L-glutamine, and 10 mM D-glucose. Manufacturer’s directions were otherwise followed for the Real-Time ATP Rate Assay Kit (Agilent Technologies).

### 2.11. Acylcarnitine Profile Analysis

Acylcarnitine profiles were determined as previously described with minor modifications [47,48,49]. Cells were seeded at 350,000 cells per well in 6-well plates in triplicate in complete DMEM. Growth medium was changed to Ham’s F12 media (Gibco, Waltham, MA, USA) supplemented with 10% FBS, 4 mM L-glutamine, and 100 IU penicillin and 100 µg/mL streptomycin (Corning Life Sciences) for 24 h. Wells were incubated with REN001 200 µM palmitic acid, 400 µM L-carnitine, and 0.4% fatty-acid-free BSA in Minimum Essential Medium (MEM; Gibco) with no supplementation. Plates were incubated in a 37 °C/5% CO_2_ incubator for 72 h. Medium was collected, cells were lysed with 250 µL of RIPA buffer for 30 min at room temperature, and the protein concentration was determined. 

Aliquots (75 µL) of medium were mixed with methanol (20 µL) containing isotope-labeled carnitine standards and the protein precipitated by the addition of absolute ethanol (905 µL) and centrifugation (13,000 rpm, 10 min). A portion of the supernatant (50 µL) was dried under a stream of nitrogen gas and the acylcarnitine butyl esters generated by reaction (60 °C for 15 min) in 100 µL of 3N HCl in butanol. Dried residues were reconstituted in acetonitrile–water (80:20) for flow injection ESI-MS-MS analysis. Analysis was performed on a triple quadrupole API4000 mass spectrometer (AB Sciex™, Framingham, MA, USA) equipped with an ExionLC™ 100 HPLC system (Shimadzu Scientific Instruments™, Columbia, MD, USA). Analyst™ (V1.6.3, AB Sciex ©2015) was utilized for data acquisition and ChemoView™ software (V2.0.3, AB Sciex ©2014) for quantitation using isotope-labeled carnitine standards. Acylcarnitine standards were purchased from Amsterdam UMC—VUmc (Amsterdam, NL, USA) and Cambridge Isotope Laboratories, Inc. (Andover, MA, USA). Acylcarnitines were measured using multiple reaction monitoring (MRM) for free carnitine (C0, *m*/*z* 218 > *m*/*z* 103) and acetylcarnitine (C2, *m*/*z* 260 > *m*/*z* 85) and Precursor Scan for precursor ions (Q1) of acylcarnitines (C3 to C18, scan range m/z 270 to 502) that generated a product ion (Q3) at *m/z* 85.

### 2.12. Statistical Analysis

Calculations were performed in Microsoft Excel. Student’s *t*-test was used to determine statistical significance in Prism GraphPad (Version 7, graphpad.com, accessed on 22 August 2022). 

## 3. Results

### 3.1. PPARδ Agonist Upregulates Genes Associated with Fatty Acid Oxidation and Mitochondrial ETC Complexes 

PPARδ agonists are known to upregulate the transcription of FAO and ETC genes [15,16,17,18,19,22,23]. Despite the well-established association between PPARδ and improved FAO in vitro, the direct target genes of PPARδ remain unclear [50]. To deduce the binding profile and target gene repertoire of PPARδ in an unbiased and genome-wide manner, we re-assessed prior PPARδ ChIP-seq data that were generated using HUVEC cells (GSE 50144) (PMID: 24721177) (Appendix A). MACS2 analysis of the ChIP-seq data identified the high enrichment of binding events (83%) that were localized to the intergenic region, with a consensus binding motif of GGTCAAAGGTCA that corresponds to PPARδ under the family and class of thyroid-hormone-receptor-related factors (NR1): nuclear receptors with C4 zinc fingers (JASPAR) (Appendix A). Given the enrichment of PPARδ binding sites being primarily localized within 5 kb downstream of the transcription start sites, such an observation supports PPARδ’s primary role as a DNA enhancer (Appendix A). Of note, functional interpretation of *cis*-regulatory regions of PPARδ binding peaks using GREAT and Cistrome GO analysis tools consistently revealed a high degree of confidence in the fatty acid metabolism pathway, which is predicted to be enriched by PPARδ target genes (Appendix A). Notably, PPARδ target genes were also found to be implicated in multiple human phenotypes that are highly reminiscent of FAO disorders, including, but not limited to, hypoglycemia, hepatic steatosis, and rhabdomyolysis (Appendix A). 

To determine whether REN001 increases transcripts associated with FAO, we quantified mRNA levels via real-time qPCR. In this manner, all control and VLCAD cell lines demonstrated statistical improvement in both FAO and ETC complexes at the transcript level (Figure 1A–F). Specifically, all treated VLCAD cell lines demonstrated the statistically significant upregulation of *ACADVL* in response to either 30 or 120 nM REN001, with the VLCAD-1 cell line demonstrating at least a two-fold increase in *ACADVL* when treated with 120 nM REN001 (Figure 1A). Similarly, *HADHA* and *HADHB*, the genes encoding for TFP subunits, also trended upwards with a small but statistically significant increase in transcripts (Figure 1B,C). Consistent with the positive effects of REN001 on FAO-associated genes, we also observed the upregulation of *ETFDHa*. To deduce the overall specificity of REN001 and demonstrate that the gene expression changes are not attributable to an off-target effect, we analyzed *UQCRC2* (Complex III) and *NDUFS2* (Complex I), genes that are not known to be targeted by PPARδ agonists, based on the ChIP-seq analysis (PMID: 24721177) (Appendix A). Indeed, qPCR analysis showed minimal changes in both *UQCRC2* and *NDUFS2*, validating the veracity of the drug target specificity for REN001 in our assay (Figure 1E,F, Appendix A). We decided to use the Control-1 cell line for all subsequent experiments as all three control cell lines had a similar increase in mRNA, with Control-1 having the largest increase in *ACADVL* (Figure 1A). 

### 3.2. Induction of Fatty Acid Oxidation Proteins 

To determine if the increase in *ACADVL* mRNA resulted in an increased in VLCAD protein with REN001 treatment, Western blotting and immunofluorescence microscopy were performed to determine VLCAD presence before and after treatment. VLCADD patient-derived fibroblast cell lines showed decreased VLCAD protein and/or enzyme activity that varied with the *ACADVL* mutation (Appendix A; Appendix A). Treatment with REN001 for 48 h increased VLCAD protein only in cell line 3, with a 2.1-fold increase when treated with 30 nM as demonstrated via Western blotting (Figure 2A). Neither of the other patient cell lines nor control cells showed significant changes in VLCAD protein signal, confirming the instability of the mutant protein translated from the upregulated mRNA. tSince PPARδ upregulates all the fatty acid β-oxidation genes, the level of TFPα and β subunits, the products of the *HADHA* and *HADAB* genes, respectively, was analyzed in patient cells. TFP is a component of the FAO/ETC macromolecular complex and interacts closely with VLCAD [4]. All cell lines had an increase in TFPα as demonstrated by Western blotting (Figure 2B). Control cells, along with patient cell lines 1 and 2, had increased TFPβ subunit across various concentrations (Figure 2B,C). VLCAD-1 had a 1.7-fold change in TFPβ subunit at various concentrations of REN001, while patient cell lines 2 and 4 did not (Figure 2C). While no statistical significance was found in comparing treated to untreated VLCADD cells in terms of protein amount (Figure 2A–C), the upregulation of the FAO genes may still be impactful. Since FAO gene expression increased, VLCAD overall may be more active, even though the total amount of protein is the same. Immunofluorescence (IF) staining of control fibroblasts, FB826, was performed for VLCAD and HADHA antigens (Appendix A). A 1.3-fold change in VLCAD was found with 30 nM REN001 via immunostaining (Appendix A). Minimal change occurred in HADHA immunostaining, also consistent with Western blotting. 

### 3.3. VLCAD Enzyme Activity 

VLCAD enzyme activity was measured in patient and control cells treated with REN001. While there were only small increases in *ACADVL* mRNA and VLCAD protein, there may still be increased VLCAD enzyme activity. Not surprisingly, untreated VLCAD-deficient patient-derived fibroblasts had significant reductions in VLCAD activity while maintaining normal levels of medium-chain acyl-CoA dehydrogenase (MCAD) activity, measured as a control (Appendix A). VLCAD-deficient cell lines had a variable response to REN001. VLCAD-1 and -3 had statistically significant increases in VLCAD activity at 60 and 120 nM concentrations, respectively. Neither VLCAD-2 nor VLCAD-4 showed increased activity. The control cell line showed a trend of increasing VLCAD enzyme activity with increasing REN001 concentration that was not statistically significant. MCAD activity for all cell lines at most drug concentrations was unchanged, though VLCAD-3 treated with 60 nM REN001 was slightly decreased (Appendix A). MCAD activity was not measured for VLCAD-4 due to a limited sample amount. 

### 3.4. FAO Flux Assay 

A whole-cell [^3^H]-oleate oxidation assay was used to measure overall flux through the fatty acid oxidation pathway and is a measure of VLCADD severity [51]. Fatty acid oxidation flux would be expected to improve with treatment as there was a small increase in VLCAD enzyme activity (Appendix A). VLCAD-2, -3, and the control cell line no significant changes in oleate oxidation following drug treatment (Figure 3). There was a trend towards an increase in flux in VLCAD-1 and VLCAD-4 suggesting minor improvement, but the change reached statistical significance only in VLCAD-4 treated with the highest concentration of REN001 (120 nM, Figure 3). A minimal or no change in activity was not surprising as the mRNA and protein had small changes. 

### 3.5. Whole-Cell Oximetry

We have previously shown that VLCADD cells show impaired oxidative phosphorylation as measured by whole-cell oximetry [52], including VLCAD-1 and -2 used in this study. Due to the small changes in mRNA and protein with REN001 treatment, we wanted to determine whether the cellular bioenergetics of REN001 VLCADD cells improved with treatment compared to untreated cells. The oxygen consumption rate (OCR) was measured via a Seahorse XFe96 Extracellular Analyzer. The basal respiration was increased compared to controls in all VLCAD-deficient cell lines, with an increase in maximum respiration and no change in spare capacity or ATP production, a pattern consistent with impaired oxidative phosphorylation and mitochondrial stress (Figure 4, Appendix A). Control cells showed decreases in all respiratory parameters with REN001 treatment, while they increased in VLCAD-deficient cell lines. VLCAD-1 and VLCAD-3 had the highest increase in basal respiration at 30 nM, while VLCAD-2 had the highest increase at 60 nM (Figure 4A, Appendix A). The control cell line decreased in basal respiration with an increase in REN001 (Figure 4A). Similarly, maximal respiration and spare respiratory capacity significantly increased across all VLCAD cell lines and significantly decreased in the control cell line (Figure 4B,C). Calculated ATP production also significantly increased in the VLCADD cell lines, with the highest increases at 30 nM or 60 nM (Figure 4D, Appendix A). VLCAD-2, the most severe VLCADD patient cell line, decreased in ATP production at 30 nM REN001 treatment (Figure 4D). VLCAD-2 is already operating at maximal ATP production and, with REN001 treatment, is hypothesized to reduce cellular stress and improve cellular bioenergetics. Thus, it is paradoxical and decreases ATP production. 

Previous cellular studies have reported variable results using bezafibrate, a pan-PPAR agonist, to improve cellular bioenergetics in FAO-deficient patient fibroblasts [32,52,53,54]. As a PPARδ agonist, which is specific to the FAO gene regulation, REN001 should theoretically have more directed action on the targets of interest in VLCADD than bezafibrate. To test this hypothesis, we treated control and VLCADD fibroblasts with 600 µM bezafibrate and measured the oxygen consumption rate (Appendix A). Control cells had decreased basal respiration and maximal respiration and ATP-linked respiration with bezafibrate treatment, while there was no statistically significant change in spare respiratory capacity (Figure 4E–H). VLCAD-1 and -3 had significantly reduced basal respiration, maximal respiration, and ATP-linked respiration with 600 µM bezafibrate (Figure 4E,F,H). VLCAD-4 had no statistically significant differences in basal respiration, maximal respiration, spare respiratory capacity, or ATP-linked respiration (Figure 4E–H). VLCAD-3 had no statistically significant differences in spare respiratory capacity. VLCAD-2 was an outlier in these experiments, with an improvement in all parameters with bezafibrate treatment (Figure 4E–H). 

### 3.6. ATP Production

An increase in ATP production can be indicative of less cellular stress and increased FAO protein due to REN001 treatment. In measuring whole-cell oximetry, there was an increase in ATP production in three VLCADD cell lines treated with REN001 (Figure 4D). This is not a direct measurement of ATP. We therefore directly measured ATP production with a real-time rate ATP assay via a Seahorse XFe96 Extracellular Analyzer. All cell lines significantly increased their total ATP production (Figure 5C). Glycolytic and mitochondrial ATP production were significantly increased across all VLCADD cell lines (Figure 5A,B).

### 3.7. Acylcarnitine Profile Analysis 

Since there was an increase in VLCAD enzyme activity and FAO flux, we then measured acylcarnitines in the growth media of REN001-treated cells. Acylcarnitines in media accumulate from cellular metabolism and a characteristic pattern including increases in long-chain saturated and unsaturated species can be detected in media from VLCAD-deficient cells, consistent with the profile seen in blood samples from patients. Reduction in these species following REN001 treatment would suggest improved VLCAD function. As expected, palmitoylcarnitine (C16) was elevated in growth medium from all of the VLCADD patient fibroblasts, except VLCAD-1, compared to controls (Appendix A). REN001 treatment did not decrease palmitoylcarntine in any VLCADD cell lines treated at the various concentrations (Appendix A). Palmitoylcarnitine significantly increased at 120 nM treatment in VLCAD-1, -3, and -4, suggesting that the higher dose of the drug is toxic (Appendix A). No change was seen in the media of VLCAD-2 cells at any drug level. Acetylcarnitine (C2) reflects levels of the acetyl-CoA end product of FAO and is typically lower in VLCAD-deficient patients and patient cells. Increased flux through FAO in patient cells could increase acetylcarnitine, though alternative metabolic pathways would utilize increased acetyl-CoA before it can accumulate. An increase in acetylcarnitine was not detected in media of the patient cell lines with REN001 treatments, with VLCAD-1 and -2 having statistically significantly decreased at 30 and 120 nM treatment (Appendix A). Control cells did not significantly change in acetylcarnitine with REN001 treatment (Appendix A). Both control and VLCADD cell lines did not statistically increase in C14:1 carnitine media levels, except VLCAD-4 (Appendix A). 

## 4. Discussion

There is no effective treatment for VLCAD deficiency. Rather, current treatment protocols rely on dietary restrictions and replenishment of the deficiency in energy using a supplemental dietary energy source such as medium-chain triglyceride oil or the newly FDA-approved heptanoic acid in the triglyceride form, triheptanoin [11,12,55,56]. However, the treatment is inadequate as episodes of rhabdomyolysis and cardiomyopathy persist. Gene therapy has been reported in mice, but has not been developed further for humans [57].

In this study, we focused on enhancing the expression of genes involved in mitochondrial bioenergetics, including FAO, using the PPARδ agonist REN001. We hypothesized that such a treatment could either directly or indirectly improve energy metabolism in cells from patients with VLCAD deficiency. PPARδ agonists’ hypothesized mechanism is by partly raising the amount of partially active mutant VLCAD protein; this will increase the overall amount of VLCAD enzyme and increase the FAO flux. Secondly, by increasing the expression of the respiratory chain genes, there is an increase in overall respiratory chain activity. It is also important to note that the current therapy for VLCAD deficiency, triheptanoin, relies on replenishing TCA cycle intermediates that may be secondarily deficient in VLCADD patients [58]. By upregulating the TCA cycle genes, this allows the TCA cycle to perform at near capacity to allow for a secondary benefit similar to that of triheptanoin. Our results demonstrated a statistically significant increase in *ACADVL* mRNA, with trends towards increases in *HADHA* and *HADHB* mRNA following treatment of patient cells with REN001 (Figure 1A–C). Indeed, there was also a small increase in protein levels of both VLCAD and TFP protein (Figure 2A–C and Appendix A) as well as cellular FAO flux in a dose-dependent fashion (Figure 3). More importantly, we demonstrated an improvement in the cellular overall bioenergetic state with REN001 treatment as measured by oxygen consumption and ATP production (Figure 4A–D and Figure 5, and Appendix A). These findings, in combination, indicate an improvement in the overall bioenergetic health of patient fibroblasts, and identify REN001′s potential as a therapeutic agent for VLCAD deficiency. 

Not surprisingly, the response of VLCAD-deficient cell lines to REN001 was variable given that each had a unique mutant *ACADVL* genotype, with variant VLCAD protein of variable stability. Similar results have been reported when treating VLCAD-deficient fibroblasts with bezafibrate, a pan-PPAR activist with considerably less delta activity than REN001 [32]. In the mentioned study, fibroblasts with the most protein-damaging mutation had minimal effects with bezafibrate treatment, including no rescue of VLCAD protein. An additional study of VLCADD fibroblasts from 33 different patients with 45 different *ACADVL* mutations similarly confirmed that bezafibrate treatment in cells with less damaging point mutations responded better than those with insertions, deletions, or frameshift mutations [53]. A similar finding was evident in our study. Fibroblasts VLCAD-2 (with the most severe predicted mutant genotype/least protein stability) exhibited only a small increase in enzyme activity and protein content with treatment, but no increase in fatty acid oxidation flux, and only minimal improvement in oxygen consumption rate. Of note, VLCADD fibroblasts containing the c.848T > C (p.Val283Ala) (VLCAD-4) and c.520G > A (p.Val174Met) (VLCAD-1) variants behaved similarly with REN001 treatment compared to cell lines containing the same mutations treated with bezafibrate [53,54]. Bezafibrate treatment restored FAO flux to 65 to 75% of control, with a 1.3- to 2.3-fold increase in VLCAD mRNA, and 2.2- to 4.8-fold increase in VLCAD activity in cell lines containing c.848T > C (p.Val283Ala). Bezafibrate treatment also increased FAO flux to 65% of control and a 1.3- to 2.3-fold increase in VLCAD mRNA expression in a homozygous c.520G > A (p.Val174Met). In VLCAD-1, 120 nM REN001 treatment did not restore FAO flux, and elicited a minimal increase in VLCAD activity, with a two-fold increase in VLCAD mRNA expression. Similarly, REN001 treatment minimally increased FAO flux and VLCAD expression, while VLCAD mRNA expression increased 3.1-fold in VLCAD-4, similar to bezafibrate treatment (Figure 1A). Our results confirm the need for an individual, mutation-specific approach for selecting appropriate drug therapies for VLCADD patients.

In considering the use of a PPARδ agonist for the treatment of patients, it is likely that dosing differences related to the relative delta effects of the drugs are significantly lower than those for the pan-PPAR agonist bezafibrate, and they are varied somewhat across cell lines. Bezafibrate had some activity at high concentrations (400–600 µM) in some previous in vitro studies, it was ineffective in others [52]. Of note, it has not shown efficacy in clinical trials in patients with a long-chain fatty acid oxidation disorder [37]. In our study, we found worsening or no change in oxygen consumption with bezafibrate treatment, even at a 20,000-fold higher concentration of bezafibrate (concentration based on previous published studies) compared to REN001, though one cell line appeared to respond minimally to both bezafibrate and REN001. Importantly, the minimal effective dose for REN001 in our study was 30 nM, more amendable to dose escalation as needed in patients. A clinical trial for resveratrol, proposed to have PPARα-γ agonist effects [59], showed no improvement in fatty acid oxidation or exercise capacity in VLCADD- or CPT2-deficient patients [60]. 

One limitation of this study is that the treatments were performed in patient-derived fibroblasts. VLCAD patients suffer from both cardiomyopathy and rhabdomyolysis due to dysfunction of heart and skeletal muscle, respectively [5,61,62,63,64]. Since fibroblasts contain fewer mitochondria compared to both, the translation of fibroblast results to patients remains to be proven. Additional experiments in other long-chain fatty acid oxidation disorders, including TFP, LCHAD, and CPT2 deficiencies, also are necessary to expand our results to them. An *ACADVL* null mouse model with no residual protein [65,66] is not ideal for testing REN001 given the lack of VLCAD protein. Rather, a point mutation in *ACADVL* generated via CRISPR/Cas technology would provide additional insight into the drug’s effect in a whole organism [67,68,69]. 

In summary, our results confirm that the PPARδ agonist REN001 is a potential treatment for VLCAD deficiency, exhibiting a positive effect on enzyme activity and cellular bioenergetics. Since results are mutation-specific, a personalized medicine approach will be necessary to assess the likelihood of utility based on their mutation status. 

## Figures and Tables

**Figure 1 cells-11-02635-f001:**
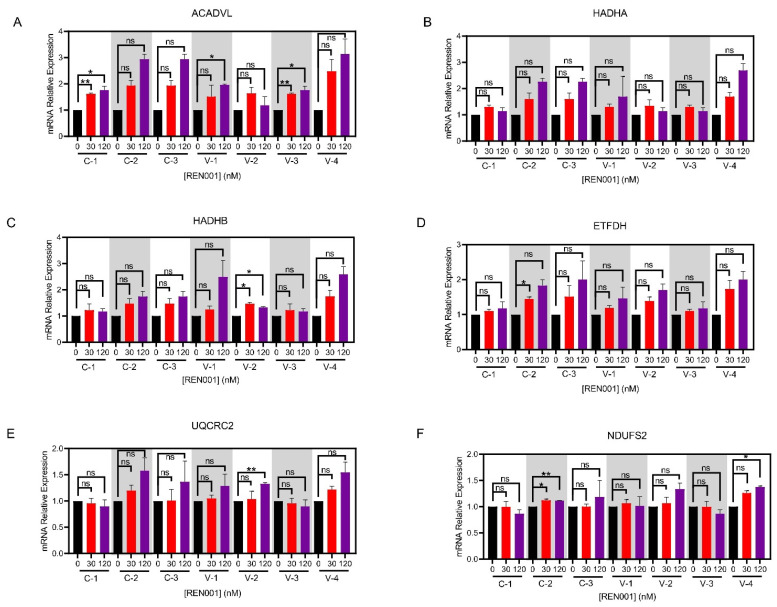
qPCR in control (C-1, 2, 3) and VLCAD-deficient (V-1, 2, 3, 4) fibroblasts (**A**–**F**) treated with REN001 at 30 or 120 nM final concentration for 48 h. Bars represent mean and standard deviations. * *p* < 0.05, ** *p* < 0.01, ns = not significant, compared to value at 0 nM treatment (n = 3 for all assays; *t*-test for unpaired samples).

**Figure 2 cells-11-02635-f002:**
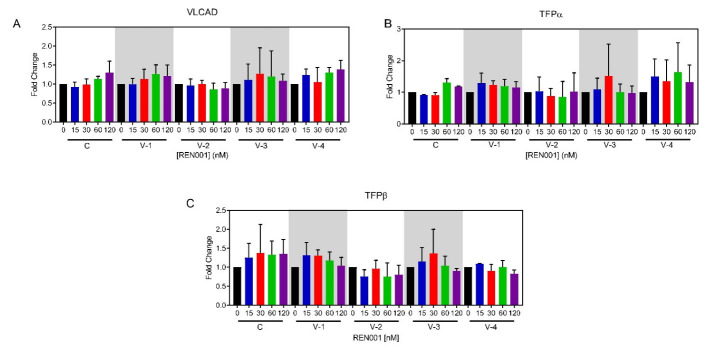
VLCAD (**A**), TFPα (**B**), and TFPβ (**C**) cellular protein content was quantified from Western blots of whole cell lysates prepared from REN001-treated fibroblasts. Data are presented as fold changes compared to values at 0 nM treatment (n = 3 for all assays). *t*-tests were performed and no statistical significance was observed.

**Figure 3 cells-11-02635-f003:**
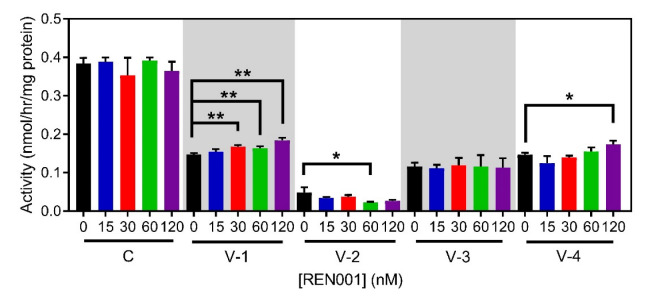
Fatty acid oxidation (FAO) flux in control and VLCAD-deficient fibroblasts treated with REN001 for 48 h. Bars represent mean and standard deviations in duplicate assays. * *p* < 0.05, ** *p* < 0.01, compared to values at 0 nM treatment (n = 3 for all assays; *t*-test for unpaired samples).

**Figure 4 cells-11-02635-f004:**
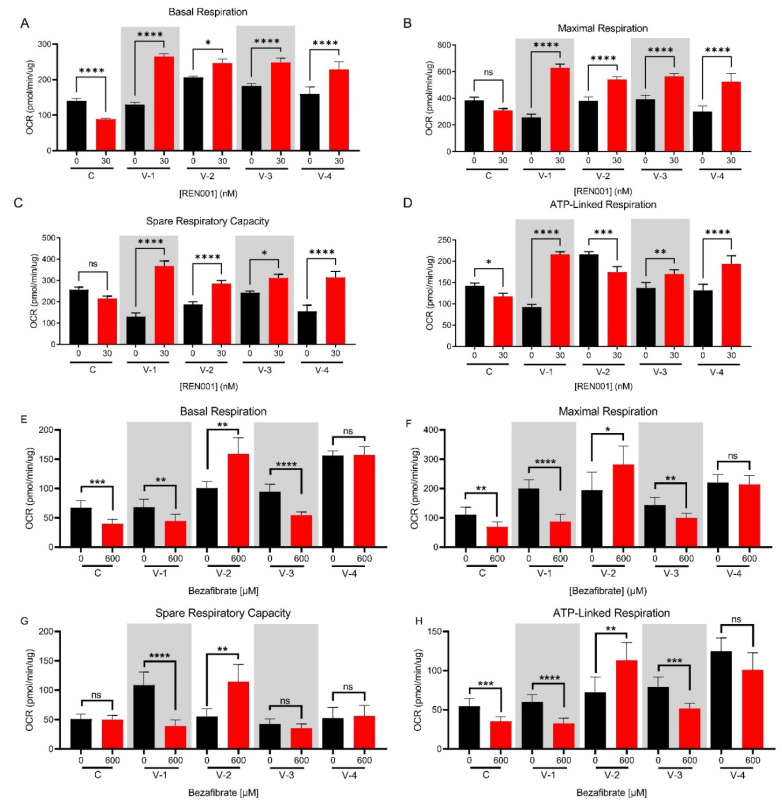
Oxygen consumption rate (OCR) of control and VLCAD-deficient cell lines treated with REN001 for 48 h (**A**–**D**). Briefly, basal respiration is the OCR of the cells under baseline conditions. Maximal respiration is the OCR measured by exposing the cells to carbonyl cyanide-4 (trifluoromethoxy) phenylhydrazone (FCCP), which is an uncoupling reagent that collapses the proton gradient and disrupts mitochondrial potential. Electron flow through the electron transport chain is disrupted and complex IV reaches the maximum OCR. Spare respiratory capacity is the cell’s ability to respond to stress via exposure to rotenone and antimycin (ROT/AA; complex I and II inhibitors, respectively). ATP production is the decrease in OCR via exposure to ATP synthase inhibitor (complex V), oligomycin, and represents the portion of basal respiration used to drive ATP production. Basal respiration (**A**), maximal respiration (**B**), spare respiratory capacity (**C**), and ATP production (**D**). Oxygen consumption rate of control and VLCAD-deficient cell lines treated with bezafibrate (BEZ) for 48 h (**E**–**H**). Basal respiration (**E**), maximal respiration (**F**), spare respiratory capacity (**G**), and ATP production (**H**). Bars represent mean and standard deviations in duplicate assays. * *p* < 0.05, ** *p* < 0.01, *** *p* <0.001, **** *p* < 0.0001, ns = no significant, compared to each cell line’s own 0 nM treatment (n = 6 for all assays; *t*-test for unpaired samples).

**Figure 5 cells-11-02635-f005:**
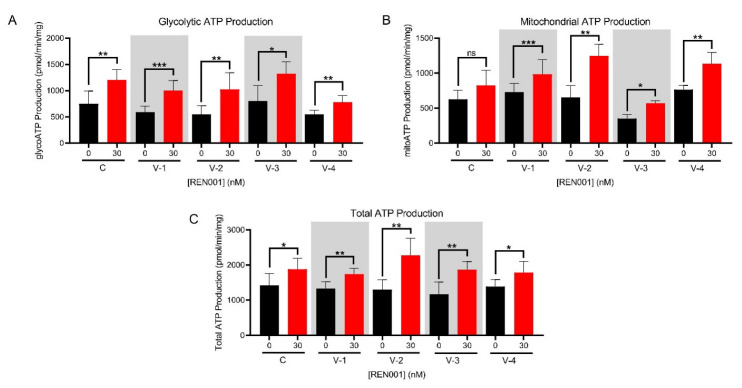
Real-time ATP production measured with Seahorse Bioanalyzer as described in the Methods in control and VLCAD-deficient fibroblasts treated with REN001 for 48 h. **A.** Glycolytic ATP production. **B.** Mitochondrial ATP. **C.** Total ATP production. Bars represent mean and standard deviations. * *p* < 0.05, ** *p* < 0.01, *** *p* < 0.001, ns = not significant, compared to values at 0 nM treatment (n = 6 for all assays; *t*-test for unpaired samples).

**Table 1 cells-11-02635-t001:** List of VLCAD-deficient cell lines used in this project with their corresponding mutations in *ACADVL* and phenotypic severity.

Cell Line	Laboratory Designation	Mutations	Phenotypic Severity
Control-1	FB826	N/A	Control
Control-2	FB549	N/A	Control
Control-3	FB902	N/A	Control
VLCAD-1	FB833	c.520G > A (p.Val174Met)/c.1825G > A (p.Glu609Lys)	Mild
VLCAD-2	FB671	c.1619T > C (p.Leu540Pro)/c.1707–1715dup (p.Asp570_Ala572dup)	Severe
VLCAD-3	FB863	c.896_898del (p.Lys299del)/c.1147C > G (p.Leu383Val)	Mild
VLCAD-4	FB782	c.848T > C (p.Val283Ala)/c.1258A > C (p.Ile420Leu)	Mild

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
