# Peer review of "Treatment of VLCAD-Deficient Patient Fibroblasts with Peroxisome Proliferator-Activated Receptor δ Agonist Improves Cellular Bioenergetics"

_cells, 2022, doi:10.3390/cells11172635_

Round 1

Reviewer 1 Report

This study  examine the possible benefits of a PPAR delta agonist as a treatment option for patients with very long chain acyl-CoA dehydrogenase deficiency (VLCADD) by lndirectly improving energy metabolism. The study uses fibroblasts  isolated from  patients with  different genotypes of  VLCAD proteins, of which one (V-2) has a more severe phenotype than the others. The effects of the PPARð ligand REN001  is studied.

Comments:

1. This is a thoroughly conducted study utilising advanced methods that is generally properly described with one exception. Fig S1 D showing human phenotypes need a more detailed description on how this figure has been prepared.

2. The study is mainly descriptive. The discussion needs  improvement as few attempts have been made to give mechanistic explanations. Four fibroblast cell lines  with differences in the mutated VLCAD gene are isolated and studied.

Using cells with different genoypes in this study should imply that differences in their response to treatment must be discussed in more depth than is the case here. Are their any relationship between genotype and response to  measured parameters? A s an example; the reasons for the differences observed between V-1 and V-2 in Fig 3 and 4 should have been penetrated.

3. The authors hypothesize that treatment with a PPARδ  ligand could either directly or indirectly improve energy  metabolism in cells from patients with VLCADD. What is the rationale behind this hypothesis? Why should  improved energy metabolism counteract the effects of  VLCAD deficiency?

4. Why Is the  PPARð ligand REN001 selected in comparison with  alternative ligands?

5. It is not obvious why data on bezafibrate effects on  OCR is included in the paper. The effects of this ligand is not discussed in relation to the given hypothesis. No attempts is made to  explain the differences  in response  between  V-1 and V-2 compared to the effects of  REN 001 (Fig 4)

6. Some data is not commented on or discussed. This relates to Fig S3 and S4 Immunofluorescence microscopy.

Minor points:

In the discussion section for each topic discussed please include reference to the relevant figure.

Legend to Fig 1: ....... VLCAD deficient (V-1, 2, 3) fibroblasts. V-4 should be included.

Author Response

This study  examine the possible benefits of a PPAR delta agonist as a treatment option for patients with very long chain acyl-CoA dehydrogenase deficiency (VLCADD) by lndirectly improving energy metabolism. The study uses fibroblasts  isolated from  patients with  different genotypes of  VLCAD proteins, of which one (V-2) has a more severe phenotype than the others. The effects of the PPARð ligand REN001  is studied. 

Comments: 

  1. This is a thoroughly conducted study utilising advanced methods that is generally properly described with one exception.Fig S1 D showing human phenotypes need a more detailed description on how this figure has been prepared.

- We have updated the materials and methods to include information on how the human phenotype figure was generated from the GREAT-GO analysis tool. We have also added some additional descriptive text to the figure legend

  1. The study is mainly descriptive. The discussion needs improvement as few attempts have been made to give mechanistic explanations. Four fibroblast cell lines with differences in the mutated VLCAD gene are isolated and studied.
  • Using cells with different genoypes in this study should imply that differences in their response to treatment must be discussed in more depth than is the case here. Are their any relationship between genotype and response to measured parameters? A s an example; the reasons for the differences observed between V-1 and V-2 in Fig 3 and 4 should have been penetrated.

We agree with the reviewer that additional mechanistic discussions will strengthen the paper and have augmented the manuscript to include them.

  1. The authors hypothesize that treatment with a PPARδ  ligand could either directly or indirectly improve energymetabolism in cells from patients with VLCADD. What is the rationale behind this hypothesis? Why should  improved energy metabolism counteract the effects of  VLCAD deficiency?
  • As a primary defect in energy metabolism, an improvement in cellular bioenergetics by definition is an improvement in the disease phenotype. We have expanded the discussion to include an explanation of the proposed mechanism of the PPARδ agonist in VLCADD patients as well as emphasized the critical nature of the physical interaction of fatty acid oxidation and respiratory chain proteins.
  1. Why Is the PPARð ligandREN001 selected in comparison with alternative ligands?

  1. It is not obvious why data on bezafibrate effects on  OCR is included in the paper.The effects of this ligand is not discussed in relation to the given hypothesis.No attempts is made to  explain the differences  in response  between  V-1 and V-2 compared to the effects of  REN 001 (Fig 4).
  • As noted in the introduction, bezafibrate, a pan-PPAR, has been proposed as a potential treatment for a variety of defects in mitochondrial energy metabolism, including fatty acid oxidation disorders. While cell studies have been contradictory for fatty acid oxidation disorders, clinical trials in patients have failed to show an effect. REN001 was selected as b example of a PPARd agonist because it is currently in clinical trials for treatment of fatty acid oxidation disorders. Thus. additional information on its mechanism of action is of clinical relevance. We have enhanced our introduction to emphasize this information

  1. Some data is not commented on or discussed. This relates to Fig S3 and S4 Immunofluorescence microscopy.
  • Thank you for pointing out this omission. Comments on immunofluorescence microscopy Fig S3 and S4 images have been added.”

Additional comments:

  • We have added V-4 to Fig 1 legend.
  • We have added the relevant reference figure in the discussion for each topic. 

Reviewer 2 Report

In this manuscript, D’Annibale et. al found that a PPARδ agonist, REN001, can rescue VLCAD at protein expression, enzyme activity and cellular function levels in several types of VLCADD fibroblasts, indicating REN001 as a potential mutation specific therapy for VLCADD. However, the data and analysis are too weak to support the conclusion at this version. Here are some questions and concerns:

1.     In Figure 1A, the authors claimed Control-1 has the largest increase in ACADVL. Does the “increase” here mean amount increase or fold increase? It makes more sense to discuss the fold increase. Control-2 seems to have the largest fold increase.

2.     In Figure 2A, did the authors perform statistical analysis, is there significant difference between different treatments? This may be the most important data to support this manuscript. If protein levels showed no significant difference, the subsequent assays make no sense.

3.     Please explain the difference among “Basal Respiration”, “Maximal Respiration”, “Spare Respiratory Capacity” and “ATP-Linked Respiration” in the main text or Figure 4 legend.

4.     In Figure 4D, why V-2 shows a decrease?

5.     In Figure S5A, why V-2 0 nM shows an OCR higher than 30 nM but lower than 60 nM? Similar thing to V-3. The increase/decrease of OCR is not gradient, please explain. 

6.     The legends for Figure 4E-H are missing.

7.     Page 2 line 89, please explain what is pan-PPAR?

8.     Page 3 lines 96-98, please explain the mechanism of REN001 detailly. How does a PPARδ agonist inhibit BNIP3? Also, please cite a paper for the proposed mechanism.

9.     There are some “comments” in the supplementary file. 

10.  Overall, the manuscript is hard to read. It is more like a work report at a lab meeting but not a paper for publication. This referee suggests to describe the results more detailly and include some analysis and discussion in the Result section.

Author Response

In this manuscript, D’Annibale et. al found that a PPARδ agonist, REN001, can rescue VLCAD at protein expression, enzyme activity and cellular function levels in several types of VLCADD fibroblasts, indicating REN001 as a potential mutation specific therapy for VLCADD. However, the data and analysis are too weak to support the conclusion at this version. Here are some questions and concerns:

  1. In Figure 1A, the authors claimed Control-1 has the largest increase in ACADVL. Does the “increase” here mean amount increase or fold increase? It makes more sense to discuss the fold increase. Control-2 seems to have the largest fold increase.

We have updated Figure 1 to be fold increase and not amount increase. While Control-2 does appear to have the largest fold increase in ACADVL, when a t test was performed, there was no statistical difference between 30 nM and 120 nM when compared to 0 nM REN001.

  1. In Figure 2A, did the authors perform statistical analysis, is there significant difference between different treatments? This may be the most important data to support this manuscript. If protein levels showed no significant difference, the subsequent assays make no sense.
  • We performed t tests comparing each cell line treatment of REN001 to its own 0 nM treatment and added a statement in the Figure 1 legend and in the results section. Even though the protein levels had no significant difference on western blotting and immunofluorescence probing, the overall increase in expression via mRNA studies indicates that the VLCAD is more active. This is measured by the ETF fluorescence reduction assay for VLCAD enzyme activity and tritium oleate release assay for FAO flux through the pathway.
  1. Please explain the difference among “Basal Respiration”, “Maximal Respiration”, “Spare Respiratory Capacity” and “ATP-Linked Respiration” in the main text or Figure 4 legend.
  • We have added a description of these parameters to the Figure 4 legend and methods.
  1. In Figure 4D, why V-2 shows a decrease?
  • V-2 is the most severe VLCADD patient cell line in this study. The cell line contains a duplication mutation and point mutation resulting in no VLCAD protein or enzyme activity. It is not clear why ATP driven respiration, an indirect measure of total cellular ATP, is reduced. However, the more pertinent direct measurement of mitochondrial ATP production (Fig 5) is increased with treatment, demonstrating improvement with treatment.
  1. In Figure S5A, why V-2 0 nM shows an OCR higher than 30 nM but lower than 60 nM? Similar thing to V-3. The increase/decrease of OCR is not gradient, please explain. 
  • We agree that this finding is unexpected and we cannot explain it. However, note that as above, overall cellular ATP improves at this concentration, suggesting that this value I merely an outlier.

  1. The legends for Figure 4E-H are missing.
  • We apologize for this oversight and have added legends for Figure 4E-H.
  1. Page 2 line 89, please explain what is pan-PPAR?
  • We have expanded our description of the PPAR promotor including the definition of a pan-PPAR agonist.
  1. Page 3 lines 96-98, please explain the mechanism of REN001 detailly. How does a PPARδ agonist inhibit BNIP3? Also, please cite a paper for the proposed mechanism.
  • We have expanded the description of REN001’s mechanism and cited a reference.
  1. There are some “comments” in the supplementary file. 
  • We apologize for failing to completely delete all internal comments intended for authors. They have been removed from the supplementary file.
  1. Overall, the manuscript is hard to read. It is more like a work report at a lab meeting but not a paper for publication. This referee suggests to describe the results more detailly and include some analysis and discussion in the Result section.
  • We thank the reviewer for this observation and the opportunity to expand the results section to include more discussion and analyses.

Round 2

Reviewer 1 Report

Revision seems appropriate

Author Response

Thank you for your time reviewing this manuscript. 

Reviewer 2 Report

The authors addressed my concerns and the quality of manuscript has been improved. 

Author Response

Thank you for your time reviewing this manuscript